# Comparison of health care resource utilization among preterm and term infants hospitalized with Human Respiratory Syncytial Virus infections: A systematic review and meta-analysis of retrospective cohort studies

**Sebastien Kenmoe**[1], **Cyprien Kengne-Nde**[2], **Abdou Fatawou Modiyinji**[1,3], **Giuseppina La Rosa**[4], **Richard Njouom**[1] *

**1** Department of Virology, Centre Pasteur of Cameroon, Yaoundé, Cameroon, **2** National AIDS Control Committee, Epidemiological Surveillance, Evaluation and Research Unit, Yaoundé, Cameroon, **3** Department of Animals Biology and Physiology, Faculty of Sciences, University of Yaoundé I, Yaoundé, Cameroon, **4** Italian National Institute of Health, Viale Regina Elena, Rome, Italy

* njouom@pasteur-yaounde.org, njouom@yahoo.com

## Abstract

### Introduction

Data on the variation in the medical resource utilization rate of Human Respiratory Syncytial Virus (HRSV) infected children by gestational age have recently been made available. This review aimed to determine whether prematurity is independently associated with the use of medical resources in hospitalized children for HRSV infections.

### Methods

We conducted this systematic review on cohort studies published on the medical resources use in preterm and full-term patients hospitalized for confirmed HRSV infections. We searched PubMed, Embase, and Global Index medicus for eligible studies. The standardized mean difference (SMD) and Risk Ratio (RR) with their 95% confidence intervals (95% CI) were estimated as summary statistics with random effects meta-analysis. The overall results were adjusted to the common confounders by stratified analyses.

### Results

A total of 14 articles (20 studies) were included. Compared to full-term, preterm hospitalized with HRSV infections had more frequent intensive care unit admission (RR = 2.6, 95% CI = 1.9–3.5), increased length of stay in hospital (SMD = 0.6, 95% CI = 0.5–0.8) and intensive care unit (SMD = 0.6, 95% CI = 0.4–0.8) and increased case fatality rate (RR = 6.9, 95% CI = 2.0–23.8). Mechanical ventilation utilization was more frequent in preterm children ≤ 2 years (RR = 15.5, 95% CI = 8.9–26.4) and those who did not receive prophylaxis against HRSV (RR = 15.9, 95% CI = 9.1–27.9)] than in full-term children. No differences were

**Data Availability Statement:** All relevant data are within the manuscript and its Supporting Information files.

**Funding:** The author(s) received no specific funding for this work.

**Competing interests:** The authors have declared that no competing interests exist.

identified in the frequency of emergency department visits, oxygen utilization, and the age at the first HRSV episode between preterm and full-term infants.

## Conclusions

Regardless of gestational age, preterm infants hospitalized for HRSV infections, especially those ≤ 2 years, have an increased frequency of use of health resources and poor outcomes compared to full-term infants. HRSV vaccine development programs for pregnant women should be accelerated.

## Clinical trials registration

Review registration PROSPERO, CRD42019124375.

## Introduction

Human Respiratory syncytial virus (HRSV) is one of the leading causes of acute lower respiratory infection in children worldwide [1–3]. Almost all children are exposed at least once to HRSV at 2 years old, with infection peak occurring between 2–6 months of life [4]. HRSV does not induce long lasting immunity and is therefore a common cause of reinfection [4]. HRSV severe infections are responsible for hospitalization, Intensive Care Unit (ICU) admission, Emergency Department (ED) visits, and the use of Mechanical Ventilation (MV) and/or oxygen supplementation registered mostly in children less than 1 year of age [5–9]. In addition to this burden the HRSV is also responsible for a high Case Fatality Rate (CFR) in resource-constrained countries [2]. HRSV infections are also responsible for a substantial economic burden on society in terms of the costs of palivizumab immunoprophylaxis, hospitalization costs, and productivity and time lost [10–12]. Several risk factors for infection, severity and mortality of HRSV infections in infants have been identified [13,14]. The most important included low birth weight, young age before or during the HRSV season, immunodeficiency, presence of other conditions (chronic lung diseases, bronchopneumonia disease . . .), multiple birth, history of atopy, tobacco exposure, male gender, breastfeeding within 2 months or less, overcrowding with school-age siblings or daycare attendance and prematurely infants [14]. Studies have shown increased risk of hospitalization in preterm (PT) compared to full-term (FT) infants with this hospitalization rate inversely correlated with decreased gestational age [15]. There is no antiviral or effective vaccine against HRSV and the management is mainly based on symptom and prevention of complications [16–19]. Palivizumab is a humanized monoclonal antibody used for preventative or prophylactic action against HRSV for the reduction of rates and duration of hospitalization related to severe HRSV infections in high-risk children [children with bronchopneumonia disease and/or children born less than 32 weeks of gestational age (wGA)] [20–22]. However, prophylaxis with palivizumab is recommended only for vulnerable children because of its high cost. The cost-effectiveness analysis for palivizumab allows regular review of palivizumab administration guidelines [23]. The 2014 guidelines issued restrictions on certain categories of infants (infants 29–35 wGA without comorbidities). Controversial data has been reported according to increased morbidity following 2014 guidelines restrictions in the administration of palivizumab [24–27]. Studies also reported an increased risk of hospitalization with HRSV in late PT (34 to <37 wGA) who however are not eligible to receive palivizumab prophylaxis [28]. To date, the synthesis of published data

comparing the morbidity of HRSV infections in PT (early and late PT) and FT infants remains to be done. These summary data from the comparison of the consequences of premature birth on HRSV infections are needed to further inform the public health stakeholders about the administration guidelines of palivizumab. The objectives of this review were to compare the use of health care resources related to HRSV infections in PT and FT patients.

## Methods

### Study design

This study was part of the HARIPI (HRSV Acute Respiratory Infections in Preterm Infants) systematic review and meta-analysis that aimed to describe the burden of HRSV in PT infants. The PRISMA guidelines were followed for the writing of this review (S1 Table) [29]. The protocol of this systematic summary was published in the PROSPERO database under registration number CRD42019124375. This systematic review is based on published studies with participants who have been disidentified and comply with research ethics standards and therefore the approval of an ethics committee was not necessary.

### Inclusion criteria

Studies reporting cohorts of patients of all ages hospitalized with confirmed HRSV infections were included. We adopted the classification mode of PT infants proposed by Engle et al. (PT, $\leq 34$ wGA and late PT, $> 34$ wGA) and a second classification defined in this review ($\leq 32$ wGA vs $> 32$ wGA) [30]. This second gestational age classification mode considered in this work was consistent with the 32 wGA threshold considered for eligibility to Palivizumab administration in children <1 year of age born between 29 and 32 wGA with other chronic pulmonary comorbidities recommended by the American Academy of Pediatrics in 2014 [22]. The diagnosis of HRSV infection was considered for viral laboratory detection conducted during the study or data recorded in databases or patient files. A second group of term patients ($\geq 37$ wGA) hospitalized with HRSV infections were the comparator group. Studies enrolling only patients with comorbidity (congenital malformations, low birth weight or immunocompromised) were excluded to reduce potential confounding effects. The frequency of health resources use was compared between the 2 groups (PT vs FT infants) by examining variables such as admission to ICU, ED visits. Prospective and retrospective cohort studies were included without linguistic, temporal or geographic restrictions.

### Search strategy

The articles published until July 22, 2019 were consulted by an electronic search in the Pubmed, Embase, and Global Index medicus databases. The research equation focused on premature birth, HRSV and health care utilization (S2 Table). References list of relevant review and included articles were screened for additional potential eligible articles.

### Study selection

We used the EndNote Reference Manager software to remove duplicates. Titles and article abstracts were independently analyzed and selected by two authors (SK and AFM). The disagreements were resolved by discussion between the authors and a third referee (RN).

### Data extraction

We (SK and AFM) extracted article related data such as: first author, year of publication, study design, recruitment period, timing of data collection, and country. We collected

participant related data such as: age at inclusion, gestational age, used of prophylaxis against HRSV, indicators of hospitalization severity such as length of stay (LOS), admission to the ICU and/or use of MV, ED visits, and HRSV detection assay used. The data collected for the use of health care resources was the total of PT and the total of FT and the number presenting the outcome for each group. The data collected for the confounding factors were the total numbers of PT and FT infant and the numbers presenting the confounding factor in each group.

## Risk of bias assessment

The included studies were independently evaluated by two authors (SK and AFM) at the methodological level using the Newcastle Ottawa Scale for non-randomised comparative studies [31] (S3 Table). Newcastle Ottawa for cohort studies is a 13-star scale that evaluates the selection of participants (6 stars), the comparability of groups (2 stars) and the outcomes (5 stars).

## Statistical analysis

We used meta-analyses for binary data (metabin) and continuous data (metacont) in the R software version 3.5.1 [32,33]. For a minimum of three studies, these functions allow to estimate all the parameters that are important for the meta-analyses (effect, heterogeneity and publication bias). This is the reason why we considered for our review, only the outcomes with at least three studies. Data from included studies (numbers, means, and standard deviation) were used to evaluate risk ratio (RR), standardised mean difference (SMD) with their 95% CIs using a random effect model [34]. We chose RR rather than OR because this parameter is indicated in measuring effect for prospective studies with binary outcomes [35]. We used the SMD rather than the mean difference because this parameter is more generalizable [36]. Means and standard deviations were estimated for studies in which they were not provided using the method described by Wan et al [37]. Differences between groups were considered statistically significant for values of p <0.05. Heterogeneity across included studies was assessed using the $I^2$ values from the Q test statistic for which a value > 50% was considered to imply heterogeneity. Subgroup analyses including gestational age, use of HRSV prophylaxis, HRSV detection assays (International Classification of Diseases (ICD) code vs laboratory detection assay), and WHO region were done to investigate the possible sources of heterogeneity. We used the symmetry of funnel plots and did the Egger test to assess the presence of publication and selective reporting bias [38]. A p-value < 0.1 was considered indicative of statistically significant publication bias. The robustness of the results was estimated through sensitivity analyses including only studies with a low risk of bias and patients ≤ 2 years. We evaluated the influence of confounding factors on overall findings by stratified analyses, including bronchopulmonary dysplasia, HRSV co-detection with other viruses or bacteria, presence of other underlying medical conditions, breastfeeding, crowding, smoke exposure, age before/during HRSV season, male gender, multiple gestation, and asthma in family history. For each potential confounding factor, we used the total numbers of PT and FT infants and the number with the confounding factor in the PT and FT group from the included studies to recalculate the p-value using the Khi-2 and Fisher exact tests. For Fisher exact or Khi-2 p-value <0.05, we considered the confounding factor to have an asymmetric distribution between PT and FT infant and symmetric if else. To appreciate the effect when the confounding factor is controlled, we conducted a sensitivity analysis with included studies with a symmetric distribution of cofounders between PT and FT infants.

## Results

### Study selection and characteristics

Our search strategy found 3361 articles through database searching and 7 through other sources and we removed 829 duplicates (Fig 1). We consulted the titles and abstracts of 2519 articles and eliminated 2444 irrelevant according to inclusion criteria of the study. We consulted the full text of 76 articles and eliminated 62 for multiple reasons presented in S4 Table. We finally included 14 articles corresponding to 20 studies whose characteristics are presented in the S5 Table [39–52]. The majority of included studies enrolled participants less than one year (9 studies) who did not receive prophylaxis against HRSV (9 studies), and were conducted in the United States (10 studies). The included studies were published between 1990 and 2018. Study participants were recruited between 1985 and 2017.

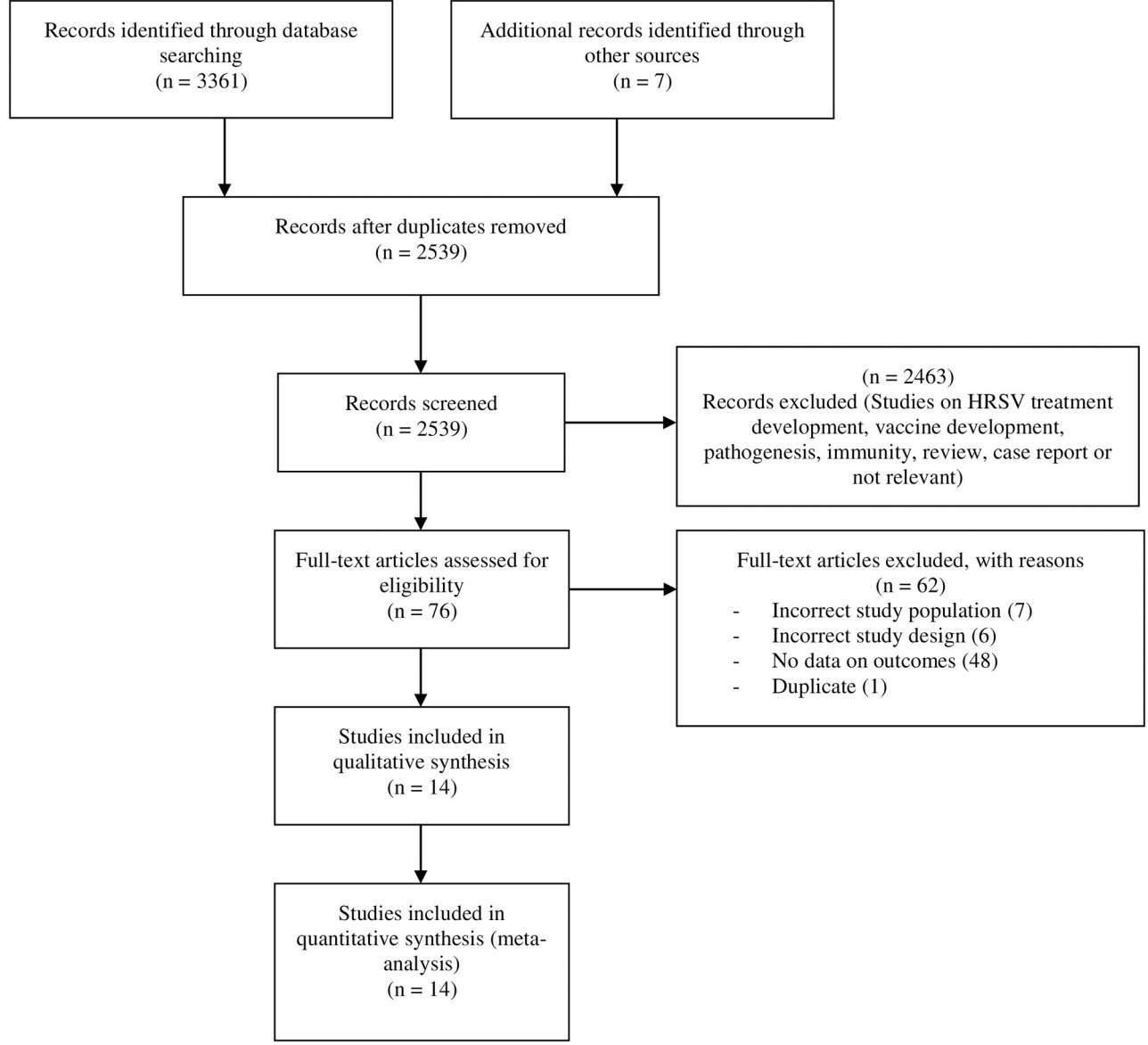

**Fig 1. Flow chart for the systematic literature search.**

### Included studies risk of bias assessment

The majority of studies had a low risk of bias (18 studies, S6 Table). Most of the included studies were population-based (15 studies) and therefore representative of the population source. All FT children were recruited from the same population as the PT infants and therefore had no selection bias. Gestational age data were collected from secure patient file and HRSV infections were confirmed by laboratory assays, ICD codes or secure recordings in almost all included studies (18 studies). The majority of included studies did not match participants on age at inclusion (17 studies). Five studies [39,43,45,46,49], however, excluded participants with comorbidities, thereby reducing the confounding factors of these parameters on the effect of prematurity on health care utilization reported by the study.

### Meta-analyses and sensitivity analyses

Fifteen studies on the association between premature birth and ICU admission shown a combined RR of 2.6 (95% CI = 1.9–3.5, $I^2$ = 83.2%) (Fig 2). The CFR was significantly higher in PT infants enrolled in six studies (RR = 6.9, 95% CI = 2.0–23.8, $I^2$ = 5.2%). Children born prematurely had LOS at hospital (SMD = 0.6, 95% CI = 0.5; 0.8, $I^2$ = 77.8%) and at ICU (SMD = 0.6, 95% CI = 0.4; 0.8, $I^2$ = 75.0%) significantly higher than FT infants (Fig 3). There was no difference for age at the time of the first episode of HRSV infection between PT and FT infants (SMD = 0.0, 95% CI = -0.0; 0.1, $I^2$ = 16.6%). The frequency of ED visits (RR = 1.1, 95% CI = 0.5–2.1, $I^2$ = 62.6%) and oxygen supplementation (RR = 1.2, 95% CI = 0.9–1.6, $I^2$ = 91.4%) were not different between PT and FT infants. There was no association in the overall analysis of the risk of MV utilization between PT and FT infants (RR = 1.9, 95% CI = 0.5–7.1, $I^2$ = 94.8). %), but sensitivity analysis with only children aged ≤ 2 years shows a significantly higher risk in PT infants (RR = 15.5, 95% CI = 8.9–26.9, $I^2$ = 0.0%) (Tables 1 and 2). Results of sensitivity analyzes that included only low-risk studies and children ≤ 2 years of age did not affect RR or 95% CI for ICU admission, ED visits, oxygen supplementation, CFR, LOS at hospital and at ICU, and age at time of index HRSV infection, suggesting the robustness of our meta-analyses.

### Heterogeneity analyses and publication bias

The studies included in the analyses of ED visits ($I^2$ = 54.0%, P = 0.114), CFR ($I^2$ = 5.2%, P = 0.304), and age at the first episode of HRSV infection ($I^2$ = 16.6%, P = 0.309) were homogeneous. Significant heterogeneity was observed in the remaining analyses. The funnel plot was asymmetric for studies involving the ED visits (P Egger = 0.027, S2 Fig). No publication bias was recorded for the remaining analyses (S1–S8 Figs).

### Subgroup analysis

The risk of admission to ICU was significantly higher in PT compared to FT in all WHO regions represented in the review (S7 and S8 Tables). An increased rate of ICU admission in PT infants was found in the Western Pacific (RR = 5.9, 95% CI = 4.6–7.5) and Europe (RR = 3.5, 95% CI = 2.9–4.2) compared to America (RR = 1.8, 95% CI = 1.4–2.2) (p < 0.001). The use of MV was significantly greater in PT children who did not receive prophylaxis against HRSV (p = 0.002). The risk of MV utilization in PT infants was significantly higher (p <0.001) in the Western Pacific (RR = 15.9, 95% CI = 9.1–27.9), in America (RR = 5.2, CI 95% = 2.4–11.7) compared to Europe (RR = 0.6, 95% CI = 0.3–1.4) where there was no association. The use of MV was statistically higher among PT compared to FT infants in the studies that identified HRSV from International Classification of Disease codes (RR = 15.5, 95% CI = 8.9–26.9)

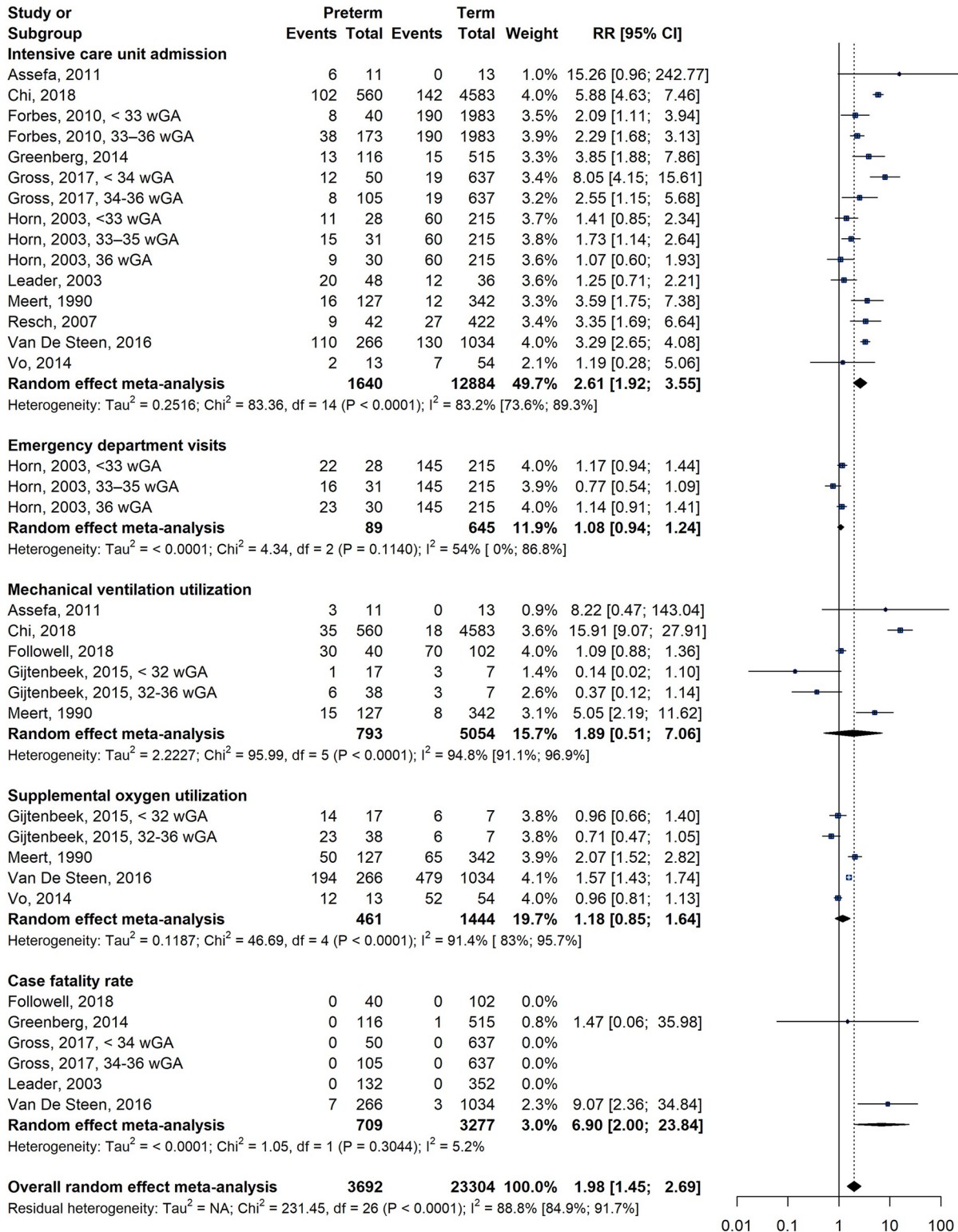

| Study or Subgroup | Preterm Events | Total | Term Events | Total | Weight | RR [95% CI] |
|---|---|---|---|---|---|---|
| **Intensive care unit admission** | | | | | | |
| Assefa, 2011 | 6 | 11 | 0 | 13 | 1.0% | 15.26 [0.96; 242.77] |
| Chi, 2018 | 102 | 560 | 142 | 4583 | 4.0% | 5.88 [4.63; 7.46] |
| Forbes, 2010, < 33 wGA | 8 | 40 | 190 | 1983 | 3.5% | 2.09 [1.11; 3.94] |
| Forbes, 2010, 33–36 wGA | 38 | 173 | 190 | 1983 | 3.9% | 2.29 [1.68; 3.13] |
| Greenberg, 2014 | 13 | 116 | 15 | 515 | 3.3% | 3.85 [1.88; 7.86] |
| Gross, 2017, < 34 wGA | 12 | 50 | 19 | 637 | 3.4% | 8.05 [4.15; 15.61] |
| Gross, 2017, 34-36 wGA | 8 | 105 | 19 | 637 | 3.2% | 2.55 [1.15; 5.68] |
| Horn, 2003, <33 wGA | 11 | 28 | 60 | 215 | 3.7% | 1.41 [0.85; 2.34] |
| Horn, 2003, 33–35 wGA | 15 | 31 | 60 | 215 | 3.8% | 1.73 [1.14; 2.64] |
| Horn, 2003, 36 wGA | 9 | 30 | 60 | 215 | 3.6% | 1.07 [0.60; 1.93] |
| Leader, 2003 | 20 | 48 | 12 | 36 | 3.6% | 1.25 [0.71; 2.21] |
| Meert, 1990 | 16 | 127 | 12 | 342 | 3.3% | 3.59 [1.75; 7.38] |
| Resch, 2007 | 9 | 42 | 27 | 422 | 3.4% | 3.35 [1.69; 6.64] |
| Van De Steen, 2016 | 110 | 266 | 130 | 1034 | 4.0% | 3.29 [2.65; 4.08] |
| Vo, 2014 | 2 | 13 | 7 | 54 | 2.1% | 1.19 [0.28; 5.06] |
| **Random effect meta-analysis** | | 1640 | | 12884 | 49.7% | 2.61 [1.92; 3.55] |

Heterogeneity: $Tau^2$ = 0.2516; $Chi^2$ = 83.36, df = 14 (P < 0.0001); $I^2$ = 83.2% [73.6%; 89.3%]

| Study or Subgroup | Preterm Events | Total | Term Events | Total | Weight | RR [95% CI] |
|---|---|---|---|---|---|---|
| **Emergency department visits** | | | | | | |
| Horn, 2003, <33 wGA | 22 | 28 | 145 | 215 | 4.0% | 1.17 [0.94; 1.44] |
| Horn, 2003, 33–35 wGA | 16 | 31 | 145 | 215 | 3.9% | 0.77 [0.54; 1.09] |
| Horn, 2003, 36 wGA | 23 | 30 | 145 | 215 | 4.0% | 1.14 [0.91; 1.41] |
| **Random effect meta-analysis** | | 89 | | 645 | 11.9% | 1.08 [0.94; 1.24] |

Heterogeneity: $Tau^2$ = < 0.0001; $Chi^2$ = 4.34, df = 2 (P = 0.1140); $I^2$ = 54% [0%; 86.8%]

| Study or Subgroup | Preterm Events | Total | Term Events | Total | Weight | RR [95% CI] |
|---|---|---|---|---|---|---|
| **Mechanical ventilation utilization** | | | | | | |
| Assefa, 2011 | 3 | 11 | 0 | 13 | 0.9% | 8.22 [0.47; 143.04] |
| Chi, 2018 | 35 | 560 | 18 | 4583 | 3.6% | 15.91 [9.07; 27.91] |
| Followell, 2018 | 30 | 40 | 70 | 102 | 4.0% | 1.09 [0.88; 1.36] |
| Gijtenbeek, 2015, < 32 wGA | 1 | 17 | 3 | 7 | 1.4% | 0.14 [0.02; 1.10] |
| Gijtenbeek, 2015, 32-36 wGA | 6 | 38 | 3 | 7 | 2.6% | 0.37 [0.12; 1.14] |
| Meert, 1990 | 15 | 127 | 8 | 342 | 3.1% | 5.05 [2.19; 11.62] |
| **Random effect meta-analysis** | | 793 | | 5054 | 15.7% | 1.89 [0.51; 7.06] |

Heterogeneity: $Tau^2$ = 2.2227; $Chi^2$ = 95.99, df = 5 (P < 0.0001); $I^2$ = 94.8% [91.1%; 96.9%]

| Study or Subgroup | Preterm Events | Total | Term Events | Total | Weight | RR [95% CI] |
|---|---|---|---|---|---|---|
| **Supplemental oxygen utilization** | | | | | | |
| Gijtenbeek, 2015, < 32 wGA | 14 | 17 | 6 | 7 | 3.8% | 0.96 [0.66; 1.40] |
| Gijtenbeek, 2015, 32-36 wGA | 23 | 38 | 6 | 7 | 3.8% | 0.71 [0.47; 1.05] |
| Meert, 1990 | 50 | 127 | 65 | 342 | 3.9% | 2.07 [1.52; 2.82] |
| Van De Steen, 2016 | 194 | 266 | 479 | 1034 | 4.1% | 1.57 [1.43; 1.74] |
| Vo, 2014 | 12 | 13 | 52 | 54 | 4.0% | 0.96 [0.81; 1.13] |
| **Random effect meta-analysis** | | 461 | | 1444 | 19.7% | 1.18 [0.85; 1.64] |

Heterogeneity: $Tau^2$ = 0.1187; $Chi^2$ = 46.69, df = 4 (P < 0.0001); $I^2$ = 91.4% [83%; 95.7%]

| Study or Subgroup | Preterm Events | Total | Term Events | Total | Weight | RR [95% CI] |
|---|---|---|---|---|---|---|
| **Case fatality rate** | | | | | | |
| Followell, 2018 | 0 | 40 | 0 | 102 | 0.0% | |
| Greenberg, 2014 | 0 | 116 | 1 | 515 | 0.8% | 1.47 [0.06; 35.98] |
| Gross, 2017, < 34 wGA | 0 | 50 | 0 | 637 | 0.0% | |
| Gross, 2017, 34-36 wGA | 0 | 105 | 0 | 637 | 0.0% | |
| Leader, 2003 | 0 | 132 | 0 | 352 | 0.0% | |
| Van De Steen, 2016 | 7 | 266 | 3 | 1034 | 2.3% | 9.07 [2.36; 34.84] |
| **Random effect meta-analysis** | | 709 | | 3277 | 3.0% | 6.90 [2.00; 23.84] |

Heterogeneity: $Tau^2$ = < 0.0001; $Chi^2$ = 1.05, df = 1 (P = 0.3044); $I^2$ = 5.2%

| Study or Subgroup | Preterm Events | Total | Term Events | Total | Weight | RR [95% CI] |
|---|---|---|---|---|---|---|
| **Overall random effect meta-analysis** | | 3692 | | 23304 | 100.0% | 1.98 [1.45; 2.69] |

Residual heterogeneity: $Tau^2$ = NA; $Chi^2$ = 231.45, df = 26 (P < 0.0001); $I^2$ = 88.8% [84.9%; 91.7%]

**Fig 2. Forest plot for the association between premature birth and risk of health care use for binary outcomes.**

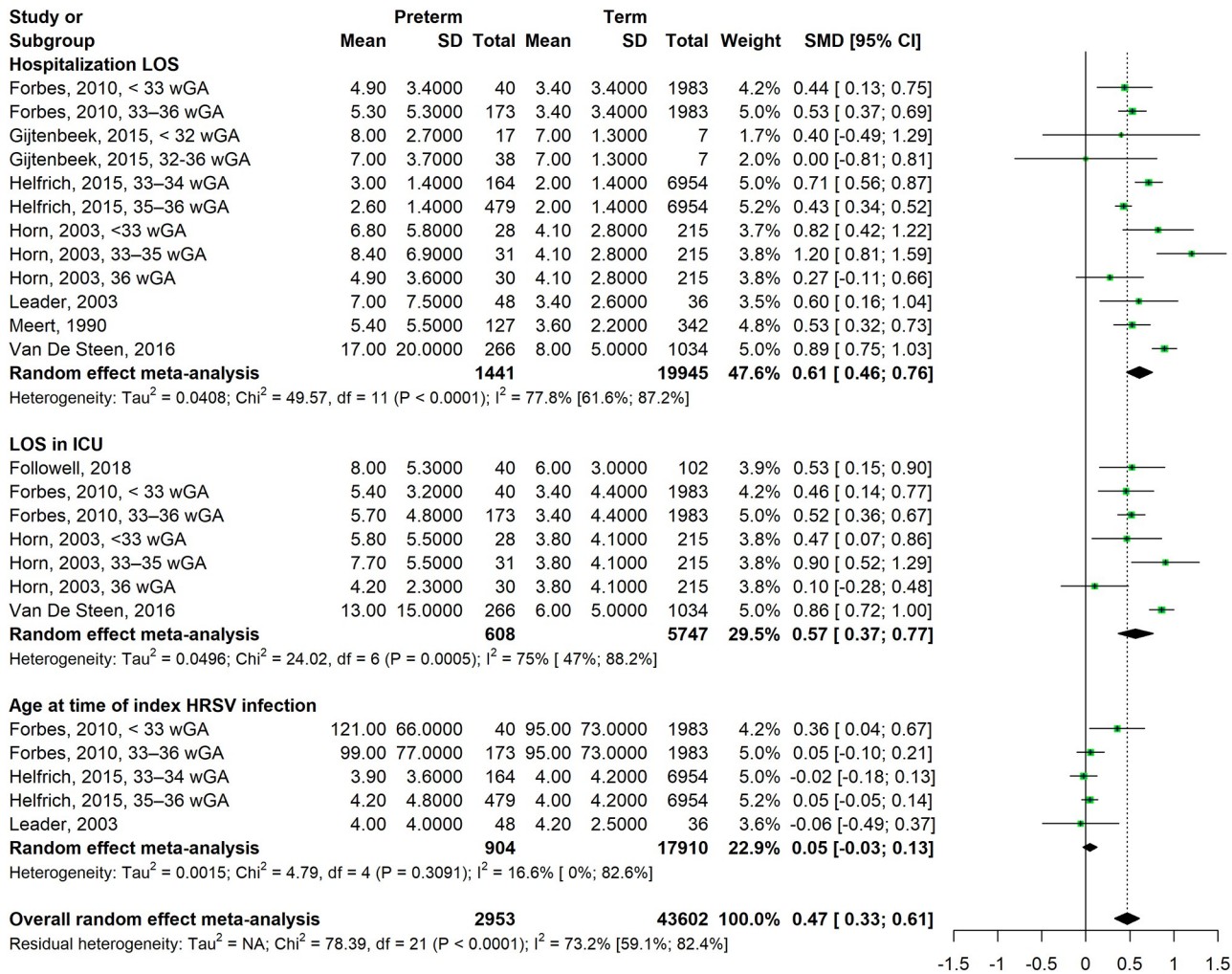

**Fig 3. Forest plot for the association between premature birth and the risk of health care use for continuous outcomes.**

compared to the authors who reported the use of laboratory assays for HRSV identification (RR = 0.3, 95% CI = 0.1–0.8) (p = 0.007). There was a significant difference in the subgroup analysis according to age at the time of the first episode of HRSV infection by gestational age of PT children (p = 0.047). Children born at a gestational age ≤ 32 wGA had an age at the time of the first episode of HRSV infection greater than FT infants (SMD = 0.4, 95% CI = 0.0, 0.7). There was no difference in the remaining subgroup analyses by gestational age of PT infants, prophylaxis against HRSV, HRSV detection assays, and WHO region.

## Confounding factors

The majority of studies that provided primary data on confounding factors had a symmetric distribution between PT and FT infants (S9 Table) including, HRSV/other virus codetection (2/2), age ≤ 3 months before/during the HRSV season (1/1), history of asthma in the family (1/1), bacterial codetection (1/1), breastfeeding < 2 months (1/1), daycare attendance during the first year (1/1), the presence of siblings of more than one year (1/1), smoking during pregnancy (1/1), twins (1/1), male gender (7/11), and bronchopulmonary dysplasia (3/4). Half of the studies with data on heart disease (3/6) and respiratory co-morbidities (1/2) were equally

**Table 1. Meta-analyses and sensitivity analyses of binary outcomes of health care resource utilization among preterm and term infants hospitalized with HRSV infections.**

| | RR (95% CI) | 95% Prediction interval | N Studies | N Preterm infants | N Term infants | H¶ (95% CI) | I²§ (95%CI) | P-value heterogeneity | P-value Egger test |
|---|---|---|---|---|---|---|---|---|---|
| **Intensive care unit admission** | | | | | | | | | |
| Overall | 2.6 [1.9–3.5] | [0.8–8.1] | 15 | 1640 | 12884 | 2.4 [1.9–3.1] | 83.2 [73.6–89.3] | < 0.001 | 0.275 |
| ≤ 2 years | 2.3 [1.6–3.2] | [0.7–7.3] | 11 | 1316 | 10846 | 2.7 [2.1–3.5] | 86.5 [77.8–91.9] | < 0.001 | 0.141 |
| Low risk of bias | 2.7 [2.0–3.7] | [0.8–8.5] | 14 | 1627 | 12830 | 2.5 [2.0–3.2] | 84.1 [74.7–90.0] | < 0.001 | 0.349 |
| **Emergency department visits** | | | | | | | | | |
| Overall | 1.1 [0.9–1.2] | [0.4–2.7] | 3 | 89 | 645 | 1.5 [1.0–2.8] | 54 [0.0–86.8] | 0.114 | 0.027 |
| ≤ 2 years | 1.1 [0.9–1.2] | [0.4–2.7] | 3 | 89 | 645 | 1.5 [1.0–2.8] | 54 [0.0–86.8] | 0.114 | 0.027 |
| Low risk of bias | 1.1 [0.9–1.2] | [0.4–2.7] | 3 | 89 | 645 | 1.5 [1.0–2.8] | 54 [0.0–86.8] | 0.114 | 0.027 |
| **Mechanical ventilation utilization** | | | | | | | | | |
| Overall | 1.9 [0.5–7.1] | [0.0–177.4] | 6 | 793 | 5054 | 4.4 [3.4–5.7] | 94.8 [91.1–96.9] | < 0.001 | 0.652 |
| ≤ 2 years | 15.5 [8.9–26.9] | NA | 2 | 571 | 4596 | 1.0 | 0.0 | 0.656 | NA |
| Low risk of bias | 2.1 [0.4–10.7] | [0.0–788.3] | 5 | 753 | 4952 | 3.5 [2.5–4.9] | 91.7 [83.5–95.8] | < 0.001 | 0.229 |
| **Supplemental oxygen utilization** | | | | | | | | | |
| Overall | 1.2 [0.9–1.6] | [0.4–4] | 5 | 461 | 1444 | 3.4 [2.4–4.8] | 91.4 [83.0–95.7] | < 0.001 | 0.424 |
| ≤ 2 years | 1.2 [0.9–1.7] | NA | 2 | 279 | 1088 | 5.1 [3.0–8.5] | 96.1 [89.0–98.6] | < 0.001 | NA |
| Low risk of bias | 1.3 [0.8–1.9] | [0.2–7.8] | 4 | 448 | 1390 | 2.9 [1.9–4.4] | 87.7 [70.9–94.8] | < 0.001 | 0.435 |
| **Case fatality rate** | | | | | | | | | |
| Overall | 6.9 [2.0–23.8] | NA | 2 | 709 | 3277 | 1.0 | 5.2 | 0.304 | NA |
| ≤ 2 years | 6.9 [2.0–23.8] | NA | 2 | 514 | 1901 | 1.0 | 5.2 | 0.304 | NA |
| Low risk of bias | 6.9 [2.0–23.8] | NA | 2 | 669 | 3175 | 1.0 | 5.2 | 0.304 | NA |

RR: Risk Ratio; N: Number; 95% CI: 95% Confidence Interval; NA: Not Applicable; LOS: Length of stay;

¶H is a measure of the extent of heterogeneity, a value of H = 1 indicates homogeneity of effects and a value of H >1indicates a potential heterogeneity of effects.

§: I² describes the proportion of total variation in study estimates that is due to heterogeneity, a value > 50% indicates presence of heterogeneity

distributed between PT and FT infants. All studies (4/4) with data on at least one underlying medical condition and passive smoking during the first year of life (1/1) had an inequitable distribution between PT and FT infants. The sensitivity analyses with symmetric distribution of each confounding factors examined revealed no major influence on the overall estimated results (S10 and S11 Tables). The increased frequency of ICU admission was lost (RR = 1.2, 95% CI = 0.3–5.1) in a study with a symmetric distribution of children with

**Table 2. Meta-analyses and sensitivity analyses of continuous outcomes of health care resource utilization among preterm and term infants hospitalized with HRSV infections.**

| | SMD (95% CI) | 95% Prediction interval | N Studies | N Preterm infants | N Term infants | H[¶] (95% CI) | I[2§] (95%CI) | P-value heterogeneity | P-value Egger test |
|---|---|---|---|---|---|---|---|---|---|
| **Hospitalization LOS** | | | | | | | | | |
| **Overall** | 0.6 [0.5; 0.8] | [0.1; 1.1] | 12 | 1441 | 19945 | 2.1 [1.6–2.8] | 77.8 [61.6–87.2] | < 0.001 | 0,763 |
| **Low risk of bias** | 0.6 [0.5; 0.8] | [0.1; 1.1] | 12 | 1441 | 19945 | 2.1 [1.6–2.8] | 77.8 [61.6–87.2] | < 0.001 | 0,763 |
| **≤ 2 years** | 0.7 [0.5; 0.9] | [0.0; 1.4] | 7 | 616 | 5681 | 2.1 [1.4–3] | 77.1 [52.1–89] | < 0.001 | 0,739 |
| **Intensive care unit LOS** | | | | | | | | | |
| **Overall** | 0.6 [0.4; 0.8] | [-0.1; 1.2] | 7 | 608 | 5747 | 2 [1.4–2.9] | 75.0 [47.0–88.2] | 0,001 | 0,262 |
| **Low risk of bias** | 0.6 [0.3; 0.8] | [-0.2; 1.3] | 6 | 568 | 5645 | 2.2 [1.5–3.2] | 78.8 [53.5–90.3] | < 0.001 | 0,333 |
| **≤ 2 years** | 0.6 [0.3; 0.8] | [-0.2; 1.3] | 6 | 568 | 5645 | 2.2 [1.5–3.2] | 78.8 [53.5–90.3] | < 0.001 | 0,333 |
| **Age at time of index HRSV infection** | | | | | | | | | |
| **Overall** | 0.0 [-0.0; 0.1] | [-0.1; 0.2] | 5 | 904 | 17910 | 1.1 [1–2.4] | 16.6 [0.0–82.6] | 0,309 | 0,622 |
| **Low risk of bias** | 0.0 [-0.0; 0.1] | [-0.1; 0.2] | 5 | 904 | 17910 | 1.1 [1–2.4] | 16.6 [0.0–82.6] | 0,309 | 0,622 |
| **≤ 2 years** | 0.1 [-0.1; 0.3] | [-1.9; 2.2] | 3 | 261 | 4002 | 1.3 [1–2.4] | 41.5 [0.0–82.2] | 0,181 | 0,820 |

SMD: Standardised Mean Difference; N: Number; 95% CI: 95% Confidence Interval; NA: Not Applicable; LOS: Length of stay;

[¶]H is a measure of the extent of heterogeneity, a value of H = 1 indicates homogeneity of effects and a value of H >1 indicates a potential heterogeneity of effects.

[§]: I[2] describes the proportion of total variation in study estimates that is due to heterogeneity, a value > 50% indicates presence of heterogeneity

bronchopneumonia dysplasia [52]. The increased hospital LOS in PT infants was lost in a study with symmetric distribution of day-care attendance 1st year (SMD = 0.0, 95% CI = -0.8; 0.8) and bronchopneumonia dysplasia (SMD = 0.4, 95% CI = -0.5; 1.3) [43].

## Discussion

This study showed that compared to FT, PT infants hospitalized for HRSV infections had a greater rate of ICU admission. Premature infants hospitalized for HRSV infections also had longer stays in the hospital and ICU than FT infants. The CFR was also higher in PT hospitalized for HRSV infections compared to FT infants. No significant differences in ED visits, age at first HRSV infection, oxygen utilization, and use of MV were observed between PT and FT infants hospitalized for HRSV infections. The use of MV, however, was significantly more common in PT infants under 2 years hospitalized for HRSV infections. Preterm infants < 32 wGA experienced the first episode of HRSV infection later than FT infants. Admission to the ICU and use of MV was significantly more common in a study of the western Pacific (Taiwan) compared to Europe and America. The risk of ICU stay was more prevalent in Europe compared to America while the risk of MV was more prevalent in America compared to Europe. Finally, we found that prophylaxis against HRSV reduced the risk of MV in the included studies.

The results obtained from this systematic review and meta-analysis are consistent with that of a previous systematic review conducted without meta-analysis with respect to increased

frequency in PT compared to FT infants for admission to ICU, LOS in hospital and ICU, and CFR. One explanation for this difference between PT and FT infants may be incomplete pulmonary and immature development of PT infants, which makes them more vulnerable to death and more severe HRSV infections, resulting in longer rates and hospital stays [53–58]. The systematic review conducted by Figueras-Aloy et al. reported that there was no difference in the use of MV and LOS in hospital between PT and FT infants. In the present study, however, a sensitivity analysis in patients aged ≤ 2 years revealed a high frequency of MV use in PT compared to FT infants. Contrary to the present systematic review which shows no difference in the frequency of oxygen supplementation in PT than in FT infants, the Figueras-Aloy et al. had reported controversial results in this regard [15]. This review highlights for the first time a similar rate of ED visit between PT and FT children. We had also shown in this review that there is no difference in age at the first episode of HRSV between PT and FT infants. This result is consistent with the review conducted by Parikh et al. on US published data and databases on patient hospitalized for HRSV infections from 2006 to 2011 [59]. While term infants are protected from HRSV infections during the first months of life by maternal antibodies [60], children born at ≤ 32 wGA are also protected by their prolonged stay in hospitals at birth [61]. This hypothesis is well supported by the subgroup analysis conducted in this work which revealed that children of ≤ 32 wGA were infected with HRSV later than FT infants. The management of HRSV infections varies considerably in countries according to economic and socio-cultural criteria, which greatly compromises the possibility of interpreting differences in the use of hospital care resources according to the WHO regions observed during this work. Several studies have shown that prophylaxis with Palivizumab helps reduce the risk of hospitalization and LOS in hospital in PT children [21,62,63]. This work has shown that children who received palivizumab did not have a greater risk of using MV.

This is the first meta-analysis to establish the association between gestational age and the use of health care resources in patients with HRSV infection. Our results were controlled for multiple confounding factors including breastfeeding, bronchopulmonary dysplasia, daycare attendance, sex, age before / during the HRSV season, tobacco exposure, family antecedent of asthma, underlying medical conditions, viral and bacterial codetections, overcrowding and multiple gestations. Apart from a study conducted in the Western Pacific west, the remaining originated from America and Europe, this represents on the one hand a weakness for this review by compromising the generalizability of findings globally and is also a strength by highlighting a gap of knowledge for other regions of the world consisting mainly of countries with limited resources that could have a particular profile and have a strong specificity in relation to the results of this work. Difference between PT with co-morbid conditions and otherwise healthy PT is also important for HRSV prophylaxis administration guidelines. The included studies in this review did not report the frequency of health care use stratified for healthy and PT with comorbidities. We were therefore not able to do this analysis as part of this review. Some included studies (5/20) used ICD codes for HRSV identification. These records are likely to be affected by inaccurate coding and may therefore incorrectly lead to the inclusion of negative HRSV participants. With the exception of the use of MV category, the group analyzes conducted during this work did not reveal any difference between these studies with the ICD codes and the studies specifying the laboratory assay used. To guarantee the robustness of our statistical analyses, we have limited our meta-analysis to outcomes available in at least 3 articles. Therefore, other relevant outcomes present in less than 3 articles including outpatient visits, antiviral or antibiotic treatment have not been included [43,49,64]. Multiple categories of subgroup analyses were represented by a single category or were not possible due to lack of data. While Grey literature or unpublished evidence would have made an important contribution to this systematic review [65], we nevertheless covered several electronic

databases including Embase which represents a database which produces unique references and we conducted robust analyses attesting that the present work is not significantly subject of publication bias [66]. Another limitation of this work is the retrospective nature of the included studies that may be affected by multiple biases, including selection bias, recall bias, and the difficulty of verifying the data collected in the archives by authors of included studies.

The results of this systematic review indicate that compared to FT hospitalized for HRSV infections, those with a history of premature birth are at increased risk of admission to ICU, leading to longer hospitalizations LOS, a higher rate of MV utilization, and higher mortality. Our results also indicate no difference between age at first episode of HRSV infection, frequency of ED visits, and oxygen supplemental utilization between PT and FT infants. In addition, according to the results of this work, compared to FT infants, there is no difference in the use of health care resources between early ($< 32/34$ wGA) and late PT infants.

Pediatricians, programme managers and policy makers should be aware that compared to FT children, early and late PT hospitalized for HRSV infections are at high risk for increased use of medical resources and poor outcomes. Special attention should be paid to PT infants $< 2$ years. Resource-limited areas such as Africa and Southeast Asia have virtually no access to or policies for the administration of HRSV prophylaxis and could thus have a particular profile for the epidemiology of HRSV infections. Additional research from these resource-limited areas is therefore important to fully understand the relationship between gestational age and HRSV infections at the global level. Additional studies of the evaluation of interventions to reduce the use of care resources by HRSV-infected premature infants are also needed.

## Supporting information

**S1 Fig. Funnel plot for publications for preterm and term children admission in intensive care unit.**
(PDF)

**S2 Fig. Funnel plot for publications for preterm and term children visit in emergency department.**
(PDF)

**S3 Fig. Funnel plot for publications for preterm and term children utilization of mechanical ventilation.**
(PDF)

**S4 Fig. Funnel plot for publications for preterm and term children utilization of supplemental oxygen.**
(PDF)

**S5 Fig. Funnel plot for publications for preterm and term children case fatality rate.**
(PDF)

**S6 Fig. Funnel plot for publications for preterm and term children hospitalization length of stay.**
(PDF)

**S7 Fig. Funnel plot for publications for preterm and term children intensive care unit length of stay.**
(PDF)

**S8 Fig. Funnel plot for publications for preterm and term children age at time of index HRSV infection.**
(PDF)

**S1 Table. Preferred reporting items for systematic reviews and meta-analyses checklist.**
(PDF)

**S2 Table. Search strategy in Embase.**
(PDF)

**S3 Table. Items for risk of bias assessment.**
(PDF)

**S4 Table. Main reasons of exclusion of eligible studies.**
(PDF)

**S5 Table. Individual characteristics of included studies.**
(PDF)

**S6 Table. Individual results of the quality assessment of the included studies using the Newcastle Ottawa Scale.**
(PDF)

**S7 Table. Subgroup analyses of binary outcomes of health care resource utilization among preterm and term infants hospitalized with HRSV infections.**
(PDF)

**S8 Table. Subgroup analyses of continuous outcomes of health care resource utilization among preterm and term infants hospitalized with HRSV infections.**
(PDF)

**S9 Table. P-value of Khi-2 and Fisher exact tests for qualitative confounding factors.**
(PDF)

**S10 Table. Sensitivity analyses of binary outcomes of the symmetrically distributed confounding factors.**
(PDF)

**S11 Table. Sensitivity analyses of continuous outcomes of the symmetrically distributed confounding factors.**
(PDF)

## Author Contributions

**Conceptualization:** Sebastien Kenmoe, Richard Njouom.

**Data curation:** Sebastien Kenmoe, Cyprien Kengne-Nde.

**Formal analysis:** Sebastien Kenmoe, Cyprien Kengne-Nde.

**Methodology:** Sebastien Kenmoe, Cyprien Kengne-Nde, Abdou Fatawou Modiyinji, Giuseppina La Rosa, Richard Njouom.

**Project administration:** Sebastien Kenmoe, Richard Njouom.

**Software:** Sebastien Kenmoe, Cyprien Kengne-Nde.

**Supervision:** Sebastien Kenmoe, Richard Njouom.

Writing – **original draft:** Sebastien Kenmoe.

Writing – **review & editing:** Sebastien Kenmoe, Cyprien Kengne-Nde, Abdou Fatawou Modiyinji, Giuseppina La Rosa, Richard Njouom.

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
