## [Decision Letter · Decision Letter 0]

7 Jan 2020

PONE-D-19-26852

Comparison of health care resource utilization among preterm and term infants hospitalized with HRSV infections: the HARIPI systematic review and meta-analysis of retrospective cohort studies.

PLOS ONE

Dear Dr Njouom,

Thank you for submitting your manuscript to PLOS ONE. After careful consideration, we feel that it has merit but does not fully meet PLOS ONE’s publication criteria as it currently stands. Therefore, we invite you to submit a revised version of the manuscript that addresses the points raised during the review process.

We would appreciate receiving your revised manuscript by Feb 21 2020 11:59PM. To enhance the reproducibility of your results, we recommend that if applicable you deposit your laboratory protocols in protocols.io, where a protocol can be assigned its own identifier (DOI) such that it can be cited independently in the future. For instructions see: http://journals.plos.org/plosone/s/submission-guidelines#loc-laboratory-protocols

We look forward to receiving your revised manuscript.

Kind regards,

Girish Chandra Bhatt, MD

Academic Editor

PLOS ONE

3. Please upload a copy of Supporting Information Figures 9 and 10 which you refer to in your text on page 15.

Reviewers' comments:

Reviewer's Responses to Questions

**Comments to the Author**

1. Is the manuscript technically sound, and do the data support the conclusions?

Reviewer #1: Yes

Reviewer #2: Yes

2. Has the statistical analysis been performed appropriately and rigorously? 

Reviewer #1: Yes

Reviewer #2: Yes

3. Have the authors made all data underlying the findings in their manuscript fully available?

Reviewer #1: Yes

Reviewer #2: Yes

4. Is the manuscript presented in an intelligible fashion and written in standard English?

Reviewer #1: Yes

Reviewer #2: Yes

5. Review Comments to the Author

Reviewer #1: Dear Authors,

I congratulate you on this very relevant piece of work.

The following are my queries/suggested modifications -

1. Were studies published in other languages apart from English also included in the analysis? (Line number 114)

2. Was there any attempt to look for grey literature/unpublished literature? (Line number 120)

3. This eligibility was defined as per convenience of available software. This would miss out assessment of outcome if a certain outcome is present in two published articles. This may be mentioned as a limitation? (Line number 149)

4. Here it is mentioned as 75 but in Figure 1, the number is 76. Please correct. (Line number 183)

5. Please give the number of such studies. (Line number 200)

6. This may not be a limitation as with a thorough search, the number of articles was limited on this particular issue, so this cannot be taken as a limitation of methodology. (Line number 358)

7. Stakeholders can be specified as hospital management , programme managers and policy makers. (Line number 373)

8. One recommendation could be to take special precaution for those < 2 years of age as that is a finding of the present study

9. This recommendation is not in line with the results obtained, so should be avoided (Line number 376)

10. In figure 1, for n=7, What are the other sources? For n=2463, a break up should be given regarding the reasons for exclusion as done for full text articles. For incorrect study type=6, does this mean incorrect study design? Then better to mention as study design.

Reviewer #2: Dear authors

Thank you very much for the laudable effort you have put in conducting this useful piece of systematic review. I have some concerns that should be addressed as outlined below

General comments

1. Please elaborte HRSV in the abstract

2. Please elaborate all the abbreviations when used first in the manuscript (e,g, wGA, PT, FT, ICU, ED, MV, ICD, etc)

3. When using the abbreviations, please follow the uniformity (either wGA or WGA , and not interchangeably)

4. I square is looking like I2

6. PLOS authors have the option to publish the peer review history of their article (what does this mean?). If published, this will include your full peer review and any attached files.

Reviewer #1: No

Reviewer #2: No

---

## [Author Response · Author response to Decision Letter 0]

10 Jan 2020

Review Comments to the Author

Reviewer #1: Dear Authors,

I congratulate you on this very relevant piece of work.

Authors: We thank the reviewer for this appreciation.

The following are my queries/suggested modifications -

1. Were studies published in other languages apart from English also included in the analysis? (Line number 114)

Authors: We plan to translate articles published in a language other than English using Google Translate. We planned this before we search for items in the various databases. However, no eligible articles were in a language other than English. All included articles are thus published in English.

2. Was there any attempt to look for grey literature/unpublished literature? (Line number 120)

Authors: We have no plans to consider gray / unpublished literature in the work. Only articles published in peer-reviewed newspapers were considered in the review. We have now reformulated line 120 to remove any confusion. 

3. This eligibility was defined as per convenience of available software. This would miss out assessment of outcome if a certain outcome is present in two published articles. This may be mentioned as a limitation? (Line number 149)

Authors: Thank you for this valuable suggestion. We added the following sentences in the discussion section: “To guarantee the robustness of our statistical analyses, we have limited our meta-analysis to outcomes available in at least 3 articles. Therefore, other relevant outcomes present in less than 3 articles including outpatient visits, antiviral or antibiotic treatment have not been included [64–66].”

4. Here it is mentioned as 75 but in Figure 1, the number is 76. Please correct. (Line number 183)

Authors: Thank you for this comment. We have revised as requested. The right number is 76.

5. Please give the number of such studies. (Line number 200)

Authors: Thank you for this comment. We have revised as requested. We added the following sentence: “Five studies [53–57], however, excluded participants with comorbidities, thereby reducing the confounding factors of these parameters on the effect of prematurity on health care utilization reported by the study.”

6. This may not be a limitation as with a thorough search, the number of articles was limited on this particular issue, so this cannot be taken as a limitation of methodology. (Line number 358)

Authors: Thank you for this comment. We have removed the limitation.

7. Stakeholders can be specified as hospital management , programme managers and policy makers. (Line number 373)

Authors: Thank you for this comment. We have revised as requested. We added the following sentence: Pediatricians and programme managers and policy makers should be aware that compared to FT children, early and late PT hospitalized for HRSV infections are at high risk for increased use of medical resources and poor outcomes.

8. One recommendation could be to take special precaution for those < 2 years of age as that is a finding of the present study

Authors: Thank you for this comment. We have revised as requested. We added the following sentence: Special attention should be paid to PT infants < 2 years.

9. This recommendation is not in line with the results obtained, so should be avoided (Line number 376)

Authors: Thank you for this comment. We have removed this recommendation.

10. In figure 1, for n=7, What are the other sources? For n=2463, a break up should be given regarding the reasons for exclusion as done for full text articles. For incorrect study type=6, does this mean incorrect study design? Then better to mention as study design.

Authors: Thank you for this comment. The other sources represent the articles we found through the screening of the list of references of the included articles and of the relevant reviews that we examined. We have now specified in figure 1 the main reasons considered for the elimination of the 2463 articles which the titles and abstracts have been screened. We have now specified the study design in Figure 1. 

Reviewer #2: Dear authors

Thank you very much for the laudable effort you have put in conducting this useful piece of systematic review. I have some concerns that should be addressed as outlined below

Authors: We thank the reviewer for this appreciation.

General comments

1. Please elaborte HRSV in the abstract

Authors: Thank you for this comment. We have revised as requested. The HRSV (Human Respiratory Syncytial Virus) is now defined at the first use.

2. Please elaborate all the abbreviations when used first in the manuscript (e,g, wGA, PT, FT, ICU, ED, MV, ICD, etc)

Authors: Thank you for this comment. We have revised as requested. The lines where all the article abbreviations were defined when they were first used are specified below.

CFR (line 59)

ICU (line 56)

ED (line 56)

PT (line 68)

FT (line 68)

wGA (line 74)

LOS (line 132)

ICD (line 160)

3. When using the abbreviations, please follow the uniformity (either wGA or WGA , and not interchangeably)

Authors: Thank you for this comment. We have revised as requested.

4. I square is looking like I2

Authors: Thank you for this comment. All the values of I square presented as I2 have now been changed to I2

---

## [Decision Letter · Decision Letter 1]

27 Jan 2020

PONE-D-19-26852R1

Comparison of health care resource utilization among preterm and term infants hospitalized with Human Respiratory Syncytial Virus infections: a systematic review and meta-analysis of retrospective cohort studies.

PLOS ONE

Dear PhD Njouom,

Thank you for submitting your manuscript to PLOS ONE. After careful consideration, we feel that it has merit but does not fully meet PLOS ONE’s publication criteria as it currently stands. Therefore, we invite you to submit a revised version of the manuscript that addresses the points raised during the review process.

We would appreciate receiving your revised manuscript by Mar 12 2020 11:59PM. To enhance the reproducibility of your results, we recommend that if applicable you deposit your laboratory protocols in protocols.io, where a protocol can be assigned its own identifier (DOI) such that it can be cited independently in the future. For instructions see: http://journals.plos.org/plosone/s/submission-guidelines#loc-laboratory-protocols

We look forward to receiving your revised manuscript.

Kind regards,

Girish Chandra Bhatt, MD

Academic Editor

PLOS ONE

Reviewers' comments:

Reviewer's Responses to Questions

**Comments to the Author**

1. If the authors have adequately addressed your comments raised in a previous round of review and you feel that this manuscript is now acceptable for publication, you may indicate that here to bypass the “Comments to the Author” section, enter your conflict of interest statement in the “Confidential to Editor” section, and submit your "Accept" recommendation.

Reviewer #1: (No Response)

Reviewer #2: All comments have been addressed

2. Is the manuscript technically sound, and do the data support the conclusions?

Reviewer #1: Yes

Reviewer #2: Yes

3. Has the statistical analysis been performed appropriately and rigorously? 

Reviewer #1: Yes

Reviewer #2: Yes

4. Have the authors made all data underlying the findings in their manuscript fully available?

Reviewer #1: Yes

Reviewer #2: Yes

5. Is the manuscript presented in an intelligible fashion and written in standard English?

Reviewer #1: Yes

Reviewer #2: Yes

6. Review Comments to the Author

Reviewer #1: Dear Authors,

Thanks for your responses. The following are my comments based on your responses -

With reference to comment made earlier under Point no. 2, since grey and unpublished literature was not searched, this should be mentioned as a limitation of the study.

With reference to comment made earlier under Point no. 10, I could not find the corrections made in Figure 1, although the authors have said that they have made corrections in Figure 1.

Reviewer #2: Dear Authors

Thank you for addressing all the comments and revising the paper accordingly

Bets wishes

7. PLOS authors have the option to publish the peer review history of their article (what does this mean?). If published, this will include your full peer review and any attached files.

Reviewer #1: No

Reviewer #2: No

---

## [Author Response · Author response to Decision Letter 1]

28 Jan 2020

Review Comments to the Author

Reviewer #1: Dear Authors,

Thanks for your responses. The following are my comments based on your responses -

With reference to comment made earlier under Point no. 2, since grey and unpublished literature was not searched, this should be mentioned as a limitation of the study.

Authors: Thank for this suggestion. We have added a sentence in the Discussion section accordingly: “While Grey literature or unpublished evidence would have made an important contribution to this systematic review [65], we nevertheless covered several electronic databases including Embase which represents a database which produces unique references and we conducted robust analyses attesting that the present work is not significantly subject of publication bias [66].”

With reference to comment made earlier under Point no. 10, I could not find the corrections made in Figure 1, although the authors have said that they have made corrections in Figure 1.

Authors: Thank you for this important reminder. Unfortunately, we provided the old version of Figure 1 after the revision. We have now provided updated Figure 1. We did not consult the full texts of the excluded articles after reading the title and summary. We cannot therefore provide details similar to that of the studies for which we have consulted the full texts, in addition this recommendation is not part of the Preferred Reporting Items for Systematic Reviews and Meta-Analyses (PRISMA) guidelines.

Reviewer #2: Dear Authors

Thank you for addressing all the comments and revising the paper accordingly

Bets wishes

We thank the reviewer for their thoughtful comments.

---

## [Editor Report · Decision Letter 2]

5 Feb 2020

Comparison of health care resource utilization among preterm and term infants hospitalized with Human Respiratory Syncytial Virus infections: a systematic review and meta-analysis of retrospective cohort studies.

PONE-D-19-26852R2

Dear Dr. Njouom,

We are pleased to inform you that your manuscript has been judged scientifically suitable for publication and will be formally accepted for publication once it complies with all outstanding technical requirements.

With kind regards,

Girish Chandra Bhatt, MD

Academic Editor

PLOS ONE

---

## [Editor Report · Acceptance letter]

7 Feb 2020

PONE-D-19-26852R2 

Comparison of health care resource utilization among preterm and term infants hospitalized with Human Respiratory Syncytial Virus infections: a systematic review and meta-analysis of retrospective cohort studies. 

Dear Dr. Njouom:

I am pleased to inform you that your manuscript has been deemed suitable for publication in PLOS ONE. Congratulations! Your manuscript is now with our production department. 

With kind regards,

on behalf of

Dr. Girish Chandra Bhatt 

Academic Editor

PLOS ONE